# Same-day versus rapid ART initiation in HIV-positive individuals presenting with symptoms of tuberculosis: Protocol for an open-label randomized non-inferiority trial in Lesotho and Malawi

Felix Gerber[1,2,3]*, Robina Semphere[4], Blaise Lukau[5], Palesa Mahlatsi[5], Timeo Mtenga[4], Tristan Lee[1,3], Maurus Kohler[1,3], Tracy Renée Glass[2,3], Alain Amstutz[1,3,6,7], Mamello Molatelle[8], Peter MacPherson[9], Nthuseng Bridgett Marake[10], Marriot Nliwasa[4,11], Irene Ayakaka[5], Rachael Burke[12]°, Niklaus Labhardt[1,3]°

1 Division of Clinical Epidemiology, Department of Clinical Research, University Hospital Basel, Basel, Switzerland, 2 Swiss Tropical and Public Health Institute, Allschwil, Switzerland, 3 University of Basel, Basel, Switzerland, 4 Helse Nord Tuberculosis Initiative, Kamuzu University of Health Sciences, Blantyre, Malawi, 5 SolidarMed Lesotho, Maseru, Lesotho, 6 Oslo Center for Biostatistics and Epidemiology, Oslo University Hospital, University of Oslo, Oslo, Norway, 7 Bristol Medical School, University of Bristol, Bristol, United Kingdom, 8 Seboche Mission Hospital Laboratory, Seboche, Lesotho, 9 School of Health and Wellbeing, University of Glasgow, Glasgow, United Kingdom, 10 Ministry of Health Lesotho, Maseru, Lesotho, 11 Malawi-Liverpool-Wellcome Trust Clinical Research Programme, Blantyre, Malawi, 12 London School of Hygiene and Tropical Medicine, London, United Kingdom

° These authors contributed equally to this work.
* felix.gerber@usb.ch

## Abstract

### Background

In absence of contraindications, same-day initiation (SDI) of antiretroviral therapy (ART) is recommended for people testing HIV-positive who are ready to start treatment. Until 2021, World Health Organization (WHO) guidelines considered the presence of TB symptoms (presumptive TB) a contraindication to SDI due to the risk of TB-immune reconstitution inflammatory syndrome (TB-IRIS). To reduce TB-IRIS risk, ART initiation was recommended to be postponed until results of TB investigations were available, and TB treatment initiated if active TB was confirmed. In 2021, the WHO guidelines changed to recommending SDI even in the presence of TB symptoms without awaiting results of TB investigations based on the assumption that TB investigations often unnecessarily delay ART initiation, increasing the risk for pre-ART attrition from care, and noting that the clinical relevance of TB-IRIS outside the central nervous system remains unclear. However, this guideline change was not based on conclusive evidence, and it remains unclear whether SDI of ART or TB test results should be prioritized in people with HIV (PWH) and presumptive TB.

### Design and methods

SaDAPT is an open-label, pragmatic, parallel, 1:1 individually randomized, non-inferiority trial comparing two strategies for the timing of ART initiation in PWH with presumptive TB

will make anonymized individual data freely available on a suitable repository, such as zenodo. org, along with publication of the study results.

**Funding:** This study is funded through a grant of the Swiss National Science Foundation (www.snf. ch; grant number 32003B_205053) awarded to NL. The funder did not and will not have a role in study design, data collection and analysis, decision to publish, or preparation of the manuscript.

**Competing interests:** The authors have declared that no competing interests exist.

("ART first" versus "TB results first"). PWH in Lesotho and Malawi, aged 12 years and older (re)initiating ART who have at least one TB symptom (cough, fever, night sweats or weight loss) and no signs of intracranial infection are eligible. After a baseline assessment, participants in the "ART first" arm will be offered SDI of ART, while those in the "TB results first" arm will be offered ART only after active TB has been confirmed or refuted. We hypothesize that the "ART first" approach is safe and non-inferior to the "TB results first" approach with regard to HIV viral suppression (<400 copies/ml) six months after enrolment. Secondary outcomes include retention in care and adverse events consistent with TB-IRIS.

## Expected outcomes

SaDAPT will provide evidence on the safety and effects of SDI of ART in PWH with presumptive TB in a pragmatic clinical trial setting.

## Trial registration

The trial has been registered on clinicaltrials.gov (NCT05452616; July 11 2022).

## Introduction

Global efforts have led to a substantial decline in HIV transmission and mortality over the past decades. A main driver of this success has been the scaling up of antiretroviral therapy (ART) care in resource-constrained settings using a public health approach, which includes the use of standardized treatment regimens with faster and simplified initiation procedures [1]. Nevertheless, HIV remains a major cause of morbidity and mortality, causing 650'000 deaths in 2021 [2]. The leading cause of mortality in people with HIV (PWH) is tuberculosis (TB) causing an estimated 187'000 deaths in PWH in 2021 [2, 3].

Since 2017, the World Health Organization (WHO) has recommended rapid ART initiation–defined as initiating ART within seven days of a positive HIV test result–to improve access to ART and to reduce pre-treatment loss to follow-up [4]. Where possible, this should include starting ART on the same day the positive HIV result is obtained. For people without clinical contraindications, rapid and same-day initiation (SDI) of ART are safe, reduce pre-treatment loss to follow up, improve retention in care, and are generally preferred by patients compared to a longer delay between HIV diagnosis and ART [5–9].

However, earlier initiation of ART, especially before the exclusion of possible opportunistic infections, may increase the risk of immune reconstitution inflammatory syndrome (IRIS) [10–12]. IRIS is an acute inflammatory condition, often presenting with worsening or new onset of symptoms of underlying opportunistic infections, caused by an exaggerated inflammatory response after restoration of immunity through ART. The risk for development of IRIS increases with lower levels of CD4 cells at baseline, the presence of opportunistic infections–especially TB–and with earlier initiation of ART if opportunistic infections are present [10, 12–15]. IRIS is most dangerous in the presence of central nervous system (CNS) infections, where SDI is specifically contraindicated [16–18].

For PWH, routine TB symptom screening is recommended using the WHO four-symptom screen (W4SS), which includes cough, fever, night sweats and weight loss [19–22]. People with at least one of the four symptoms are defined as having presumptive TB and should have TB tests to refute or confirm active TB disease. The prevalence of TB symptoms among PWH not taking ART was estimated at 71% in a systematic review, although with considerable variation

between studies [20]. Studies in Kenya and South Africa have shown that sputum tests confirmed active TB in only 5%-19% of people who had TB symptoms when presenting at HIV services before starting ART [23–25].

Prior to 2021, WHO guidelines considered the presence of TB symptoms as a contraindication to SDI and recommended deferral of ART initiation until active TB was ruled out or TB treatment was initiated [26]. In practice, because TB symptoms are so common and confirming or refuting TB within one day is often not possible in routine care, this recommendation was a considerable barrier to SDI for a significant proportion of PWH. Whilst delaying ART until TB tests results are available might be clinically reasonable in optimal settings, it may cause harm in routine practice as delays between HIV diagnosis and ART initiation are associated with attrition from care, pre-treatment mortality and substantial out-of-pocket costs for patients, especially in remote areas where walking distances or transport costs to clinics are high [27–29]. In these settings, offering SDI to PWH with presumptive TB while TB tests are concurrently done may be a valuable strategy to engage and retain more patients in care and thus improve overall clinical outcomes. Based on this reasoning, WHO included the clinical consideration to start ART in PWH with presumptive TB, but no signs of CNS infection, while rapidly investigating for TB in their guidelines since 2021 [19, 30]. This recommendation was issued despite a systematic review informing these guidelines concluding that "there is insufficient evidence about whether presence of TB symptoms should lead to ART start being deferred or not" because the effects on key health outcomes, such as mortality, TB and HIV outcomes had not been investigated [31].

The objective of the SaDAPT trial is to address this evidence gap by assessing the effect of SDI of ART ("ART first") versus rapid ART initiation after TB results are available ("TB results first") on HIV viral suppression six months after enrolment, and on other relevant health outcomes including mortality, serious adverse events (SAEs), and adverse events consistent with TB-IRIS (adverse events of special interest [AESI]), among PWH with presumptive TB and no signs of CNS infection.

## Methods

### Design and setting

SaDAPT is an open-label, pragmatic, parallel, 1:1 individually randomized, non-inferiority trial conducted at primary care clinics in urban Blantyre, Malawi, and in primary and secondary level facilities in five districts in Lesotho (urban and rural). HIV prevalence is estimated to be 21% in Lesotho and 12% in Blantyre, Malawi [2, 32]. TB incidence is 650 per 100'000 person-years in Lesotho and TB prevalence has recently been estimated at 150 per 100'000 adults in Blantyre [2, 33]. Both Malawi and Lesotho have a public health approach [1, 4] to ART delivery with HIV care being task-shifted to nurses and to lower cadres of healthcare staff.

### Eligibility criteria and enrolment

The SaDAPT trial enrolls consenting PWH, 12 years or older, not taking ART (either ART naïve or interruption for 90 days or more), who have at least one TB symptom according to the W4SS [20]. For ART-naïve and ART experienced people after relevant treatment interruption, the same risks and recommendations apply regarding ART initiation and management of possible opportunistic infections. The question of whether ART should be (re)initiated on the same day or whether TB should be investigated first is of equal relevance to both groups. Enrolling both, ART naïve as well as PWH after treatment interruption of 90 days or more will generate the evidence that is relevant to routine primary healthcare. Routine staff at study facilities refer PWH newly diagnosed or after ART interruption of at least 90 days to study staff for

| TIMEPOINT | 0 Screening enrolment | TB results review | 28 days Register review | Week 26 (22-30) Study visit | 30 week Register review | Weeks 31-40 Tracing |
|---|---|---|---|---|---|---|
| Eligibility screening | X | | | | | |
| Informed consent | X | | | | | |
| Randomization and arm allocation | X | | | | | |
| **INTERVENTION:** | | | | | | |
| ART initiation ("ART first arm") | X | | | | | |
| ART initiation ("TB results first arm") | | X | | | | |
| **ASSESSMENTS:** | | | | | | |
| Sociodemographic characteristics ascertained | X | | | | | |
| Brief clinical assessment | X | | | X | | |
| CD4 cell count | X | | | | | |
| CrAg test if CD4 <200 cells/ml or clinical indication | X | | | | | |
| Start TB tests | X | | | | | |
| Surveillance of SAEs, AESIs, active TB disease | X | X | | X | | |
| HIV Viral Load | | | | X | | (X) |
| ART adherence and refills since enrolment | | | | X | | |
| Review of routine registers to check for missing data | | | X | | X | |

**Fig 1. SPIRIT schedule of enrolment, interventions, and assessments.** Timing of the TB results review is depending on local routine procedures. Tracing only applies to participants without documented HIV viral load between week 22 and 30. HIV viral load will be measured during tracing if participant was found and no viral load result between week 22 and 30 became available. SAE: Serious Adverse Event, AESI: Adverse Event of Special Interest, CrAg: Cryptococcal antigen test.

eligibility screening. After obtaining oral screening consent, eligibility screening is done using a questionnaire in the electronic data collection tool (Research Electronic Data Capture [RED-Cap] platform [34]). Individuals with a medical condition requiring admission to hospital or referral to a higher-level health facility at enrolment, having symptoms or clinical signs of CNS infection (neck rigidity, severe headache, confusion), not planning to receive follow-up care at the study facility, reporting to be pregnant, or to be taking TB treatment, TB preventive therapy (TPT), or treatment for cryptococcal meningitis are excluded (Fig 1). In participants with a CD4 cell count below 200 cells/mm$^3$ or participants who are severely wasted or unable to walk unaided without CD4 results available at the time of screening, a serum cryptococcal antigen (CrAg) test is performed and those with a positive or missing CrAg result are excluded from the trial. After screening, eligible participants are approached for written informed consent. For adolescents below 18 years, a guardian consent is asked in addition to the adolescent's assent. Illiterate individuals give consent with a thumbprint countersigned by a witness not involved in the study.

## Baseline assessment

At baseline, we collect demographic, socioeconomic and clinical information on HIV and ART history, document CD4 cell count and, if possible, take a first TB sputum sample for analysis by nucleic acid amplification test (NAAT, Xpert® MTB/RIF or Xpert® MTB/RIF Ultra [Cepheid, USA]). Other investigations (for example serum C-reactive protein (CRP), chest X-ray, urine lipoarabinomannan assay (LAM) or additional sputum analysis), ART counselling, and prescribing and dispensing of ART, TB preventive treatment (TPT) and cotrimoxazole preventive treatment (CPT) are done at the discretion of routine staff providing ART care. At present, in Malawi, urine LAM is available but rarely used, while chest X-ray and CRP are unavailable at the study sites. At the study sites in Lesotho, urine LAM is sometimes available but rarely used, CRP is not available, and chest X-ray is available most of the time. NAATs are routinely available at all study sites with varying turnaround times. We aim to have at least one sputum NAAT result documented for every participant. However, if NAAT is not available, active TB may be confirmed or refuted based upon clinical and/or radiological findings as per

local routine practices to avoid delay of diagnostic and therapeutic procedures through trial participation.

## Randomization and blinding

After assessment of baseline characteristics and initiation of TB investigations, participants are randomized using block randomization with varying block sizes of four and six using the ralloc command in Stata, version 16.1 [35], stratified by country, based on randomization lists that were generated by an independent statistician and uploaded into the Research Electronic Data Capture (REDCap) [34] study database. Site investigators allocate participants to the interventions using the REDCap randomization module on password protected study tablets.

Because of the nature of the study and the interventions offered, it is not feasible to blind participants or site investigators to group allocation. Viral load results, the primary endpoint, are measured by laboratory staff who are not directly involved in the study and remain blinded to trial arm allocation. The endpoint of adverse events consistent with TB-IRIS is assessed through review of clinical reports by an independent expert panel blinded to arm allocation, as the assessment of this endpoint is more subjective and therefore vulnerable to bias if allocation was known.

## Trial intervention

The trial tests two different strategies for the timing of ART initiation relative to completion of initial TB investigations ("TB results first" versus "ART first"). Other than the arm-specific timing of ART initiation relative to completion of initial TB investigations, the trial participation has no influence on the clinical management of participants. It is expected that in both countries routine clinical staff will mostly adhere to the current local guidelines [36–38] regarding ART initiation and management of potential opportunistic infections. While non-adherence to guidelines by routine clinical staff will be documented in the trial data base and reported, it will not be considered a study protocol deviation.

## TB results first arm

For participants in the "TB results first" arm, ART initiation is deferred until TB investigations are complete and active TB has been refuted or confirmed, with the aim to start ART within seven days after first presentation (i.e. rapid ART initiation, Fig 2). The duration until investigations for active TB are complete, depends on individual clinical presentation, local routine procedures, and availability of TB examinations. For most participants, TB investigations are considered complete after a NAAT sputum result is obtained. However, in some cases, further examinations after a negative NAAT result are required, while others might be diagnosed based on clinical and/or radiological findings without NAAT result. For some participants, for example those presenting in the morning hours at a study site with quick NAAT return times, the TB investigations are completed on the day of presentation and ART may be initiated on the same day after results are available. For other participants, TB investigations are more time consuming, and follow-up visit on another day may be required to discuss results with the participant and initiate treatment. The study team does not influence duration or interpretation of TB investigations, but documents the routine procedures and results.

## ART first arm

For participants in the "ART first" arm, SDI of ART is offered regardless of whether TB investigations have been completed or not (Fig 2). Routine clinical staff is responsible for communication of TB test results and initiation of TB treatment in case of positive results.

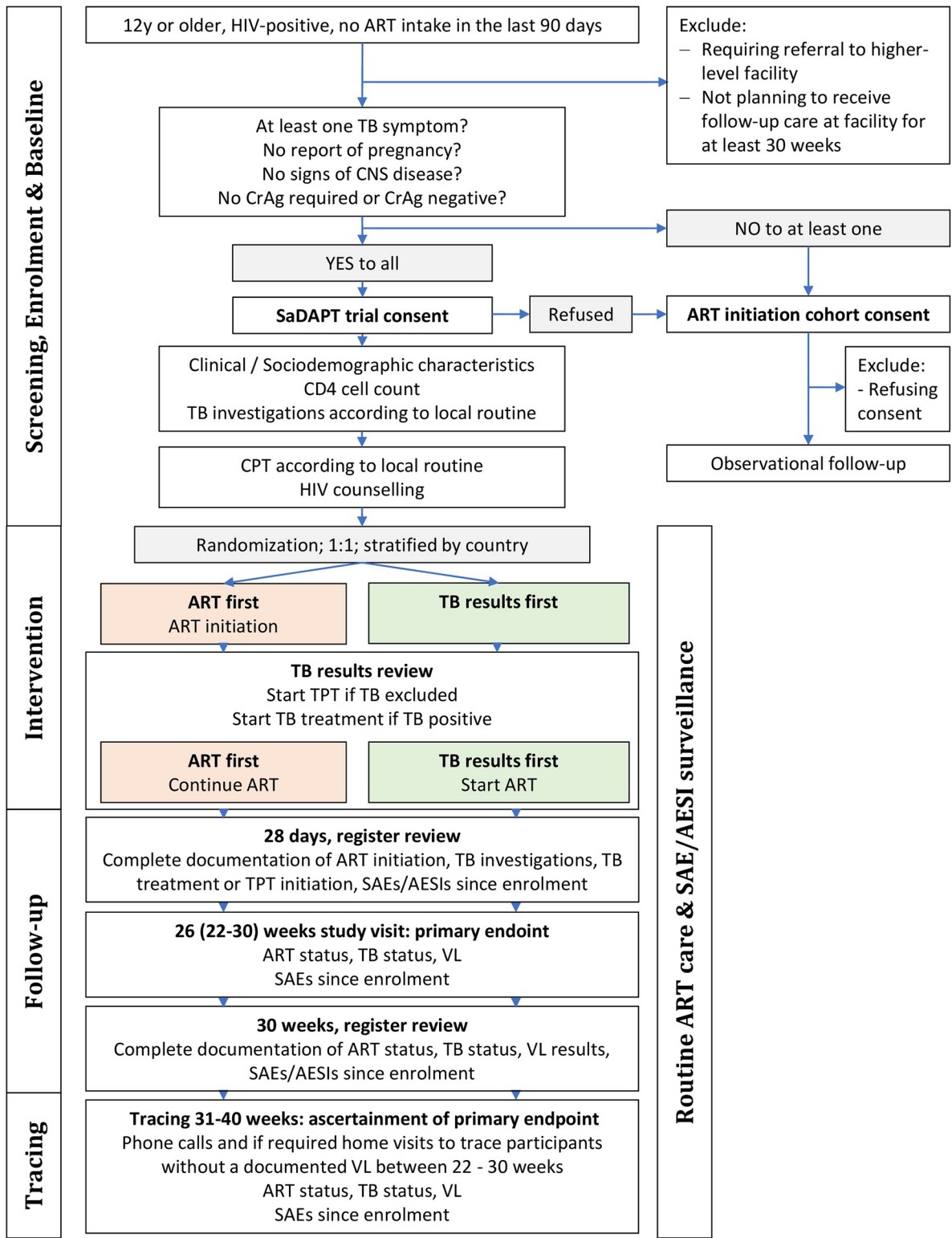

**Fig 2. Consort chart of the SaDAPT trial.** CNS: Central nervous system, CrAg: Cryptococcal antigen, CPT: Cotrimoxazole preventive therapy, TPT: Tuberculosis preventive therapy, SAE: Serious adverse event, AESI: Adverse event of special interest, VL: viral load.

## Follow-up and endpoint assessment

The primary outcome is HIV viral suppression (<400 copies/mL) six months (measurement window 22 to 40 weeks) after enrolment (Table 1). VL results are obtained from laboratory reports at the study facility or a referral facility (in case of transfer), or from dried blood spot (DBS) samples for participants without documented clinic visit but found during home visit tracing.

Secondary endpoints are initiation of ART by 7 and 28 days after enrolment, retention and engagement in ART care, disengagement from ART care and loss to follow-up six months after enrolment, non-traumatic mortality, serious adverse events (SAEs), adverse events consistent with TB-IRIS (adverse events of special interest [AESIs]), and incidence of active TB disease (see Table 1 for exact definitions).

Participants are instructed to contact the study team if they become unwell or are admitted to hospital over the course of the study. Routine staff in ART clinics are referring participants to trial staff for study documentation in case of relevant health problems during the follow-up period (passive surveillance for SAEs and AESIs). The trial team does not intervene in routine care, but documents relevant events and therapeutic procedures in the trial database, for example the date and circumstances of ART, TPT or TB treatment initiation. There are no trial-specific scheduled visits before the endpoint window and the site investigators are not tracing participants prior to 30 weeks from enrolment.

Twenty-eight days after enrolment, a review of routine registers is conducted by the study team to ensure that relevant data, including information on ART initiation, TB tests and TPT and TB treatment initiation, are fully captured in the trial database.

The final study visit (22–30 weeks after enrolment) is scheduled together with the routine 6-month ART refill and viral load measurement visit, where possible. In addition to routine

**Table 1. Overview of endpoints.**

**Primary endpoint:**

• HIV viral suppression <400 copies/mL 26 (22–40) weeks after enrolment (obtained from laboratory reports at study facility or referral facility (in case of transfer), or from dried blood spot (DBS) sample for participants found during home visit tracing)

**Secondary endpoints:**

• Retention in care 26 (22–30) weeks after enrolment, defined as having a documented clinic visit between 22 and 30 weeks after enrolment

• Engagement in care 26 (22–30) weeks after enrolment, defined as reporting regular ART intake, between 22 and 30 weeks after enrolment

• Disengagement from care 26 (22–30) weeks after enrolment, defined as not being engaged in care but reached through patient tracing

• Lost to follow-up 26 (22–30) weeks after enrolment, defined as not being retained in care and not reached through tracing

• Non-traumatic mortality, incidence of SAEs and of adverse events consistent with TB-IRIS (AESIs) during the first 30 weeks after enrolment

• Incidence of TB disease (microbiologically confirmed and/or clinical diagnosis) during the first 30 weeks after enrolment, defined as any TB diagnosis after enrolment not classified as prevalent TB at enrolment

• ART initiation by 7 and 28 days after enrolment

**Exploratory endpoints:**

• Clinical characteristics of participants with non-traumatic hospitalizations and deaths in PWH with presumptive TB

• Prevalence of active TB diagnosed at enrolment, defined as TB diagnosed clinically or microbiologically through the TB investigations initiated at enrolment and completed up to a maximum of 28 days after enrolment.

• HIV viral suppression at 26 (22–40) weeks using different thresholds (<40 copies/mL; <100 copies/mL; <1000 copies/mL)

SAE: Serious Adverse Event; AESI: Adverse Event of Special Interest

adherence counseling, ART refill and viral load measurement, site investigators take a targeted clinical history including inquiry about ART and TB status, current health status, and possible SAEs since enrolment. For all participants, a final clinic register review is conducted 30 weeks after enrolment to ensure that all relevant routinely collected data since enrolment (SAEs, AESIs, ART status, TB treatment initiations, viral load) are documented in the trial database covering the full follow-up time of 30 weeks.

After week 30, participants without a documented viral load measurement between week 22 and 30 after enrolment are traced to ascertain their outcome. Viral load results obtained during the tracing are considered for the primary endpoint assessment up to 40 weeks after enrolment. If no viral load result is available after week 40, the participant is considered as not having reached the endpoint of viral suppression six months after enrolment. A minimum of three phone calls on different days to all contact numbers provided by the participants at enrolment are conducted. All information obtained during tracing, for example whether a participant is still alive or not, is documented. If a participant can be contacted via phone, an appointment at the study facility for endpoint assessments and re-linkage to care is given. In case of unsuccessful tracing via phone, a home visit is conducted. Participants found at home are given an appointment at the study facility and a dry blood spot are collected for viral load measurements. For participants with a reported transfer to another facility, viral load results are requested from the transfer facility. If a participant has died, the circumstances of death (in particular whether death was due to trauma and whether TB was diagnosed prior to death) is recorded.

## Statistical analysis and sample size

The primary analysis is the comparison of the proportion of participants with HIV viral suppression (<400 copies/ml) between the two study arms. For this analysis, we will use a modified poisson regression to estimate the risk difference and risk ratios with their corresponding 95% confidence intervals based on a robust standard errors. These models will be adjusted for country (the randomization stratification factor) and for baseline characteristics found to be unbalanced between study arms [39]. For the non-inferiority comparison, a confidence interval approach will be used. If the lower bound of the 95% confidence interval for the absolute risk difference does not include the non-inferiority margin of 10%, then the intervention will be considered non-inferior. The non-inferiority margin of 10% was chosen based on clinical reasoning, guideline recommendations and previous studies in the field. The US Food and Drug Administration recommends a margin of 10–12% for comparing the anchor drugs in treatment-naïve patients in HIV treatment trials [40]. This is also the margin that was used by the majority of HIV trials in the past [41]. Most importantly, a margin of 10% represents a clinically reasonable minimal effect that should be preserved in order to recommend same-day ART initiation as a safe and effective alternative to delaying ART initiation until TB results are available considering its advantages of simplifying initiation procedures for providers and clients and of reducing pre-ART loss to follow-up. The primary analysis will be done in a modified intention to treat population (mITT), including everyone randomized except participants found to have been ineligible for the trial only after randomization and in a per-protocol (PP) population including all participants who completed trial procedures without a major deviation [42]. If the "ART first" arm is found to be non-inferior to the "TB results first" arm in both the mITT and PP sets, then we will assess for superiority of the "ART first" approach in the mITT set. We plan to conduct a subgroup analysis of the primary outcome stratified by ART naïve versus ART experienced participants.

Given the expected proportion of participants with viral suppression six months after enrolment in the "TB results first" arm of 75% and the non-inferiority margin of 10%, a sample size

of 295 participants per arm is required to have 80% power to demonstrate non-inferiority. Individuals found to be ineligible post-randomization will be replaced.

The secondary endpoints of ART initiation by 7 and 28 days after enrolment, retention and engagement in care, disengagement from care, lost to follow-up, SAEs, AESIs, and non-traumatic deaths will be assessed for superiority in the mITT population. Secondary endpoints will be analyzed with modified poisson regression models adjusted for country, and results will be presented as risk differences, risk ratios and 95% confidence intervals if events allow, and otherwise be summarized descriptively. HIV viral suppression at different thresholds ($<40$, $<100$, and $<1000$ copies/mL) will be analyzed as an explorative sensitivity analysis of the primary outcome with an emphasis on effect estimates and confidence intervals. Further details will be descibed in the statistical analysis plan.

## Data management and monitoring

Data is collected on password-protected tablets, using a validated installation of the REDCap platform [34] hosted at the University Hospital Basel, Switzerland. Data is reviewed regularly by central and local data managers. Queries regarding inconsistencies and missing data are raised within the electronic data capture system. REDCap includes a computer-generated time stamped audit trail. Study participants are assigned a unique participant number, which is used to identify participants in the electronic case report forms and in all data exports.

The Swiss Tropical and Public Health Institute's Clinical Operations Unit is responsible for trial monitoring in Lesotho, and the Malawi-Liverpool-Wellcome Programme Clinical Research Support Unit in Malawi. The monitoring entails a systematic examination of study related activities and documents.

A data safety and monitoring board consisting of three independent experts will review SAE and AESI reports and provide recommendations to the study team whether to pause, stop, amend, or continue the trial to ensure participant safety and integrity of the trial data. No formal interim analysis will be conducted.

## Ethical considerations

The trial has been approved by the National Health Research and Ethics Committee (NH-REC) of Lesotho (ID: 102–2022, approved October 02, 2022), the College of Medicine Research and Ethics Committee (COMREC) of Malawi (ID: P.04/22/3612, approved June 06, 2022) and the "Ethikkommission Nordwest- und Zentralschweiz" (EKNZ) in Switzerland (ID: AO_2022_00031, approved June 30, 2022). Substantial changes to the study protocol and relevant study documents will be submitted to the involved ethics committees for approval before implementation and information will be updated on the clinical trial registry.

The tested strategies for the timing of ART initiation relative to completion of TB investigations in PWH with presumptive TB have the following risks and benefits: In the "ART first" arm, the risk of TB-IRIS is increased due to the earlier initiation of ART [13] whereas the risk of pre-ART attrition from care and thereby of AIDS-related complications, is reduced [7]. The opposite applies to the "TB results first" arm. For an in-depth risk-benefit assessment, we refer to the current WHO HIV guidelines [19] and the corresponding systematic review [31]. The rationale for this trial is the uncertainty about which of the strategies entails a more favorable overall risk-benefit profile.

## Sub study: ART initiation cohort

In parallel to the SaDAPT trial, we are recruiting participants aged 12 years and older, (re)initiating ART at the study sites but not eligible for the SaDAPT trial into an observational ART

initiation cohort. This cohort shall allow to better understand the SaDAPT trial's external validity through description of baseline characteristics and outcomes at the same clinics but outside a trial setting. The specific objectives of the ART initiation cohort are to provide a description of baseline characteristics and outcomes of PWH (re)initiating ART in primary and secondary health facilities in Malawi and Lesotho and to allow for an informal comparison between PWH (re)initiating ART with and without presumptive TB according to the W4SS [20]. The sensitivity and specificity of the W4SS for TB case finding in PWH is limited and a considerable proportion of PWH are known to have subclinical TB disease before initiating ART [20, 43, 44]. It is unclear how relevant the presence or absence of TB symptoms are for clinical management and outcomes, including occurrence of active TB [45]. The ART initiation cohort shall allow understanding if, and if so, to what extent characteristics and outcomes among PWH with and without TB symptoms who are (re)initiating ART differ. The ART initiation cohort will provide insights into the current management and outcomes of PWH (re) initiating ART in Malawi and Lesotho beyond the narrow question about timing of ART initiation in presumptive TB patients addressed by the SaDAPT trial.

Only patients requiring referral to a higher-level facility at the time of screening, patients planning to move out of the facility's catchment area within the next six months or those not consenting are excluded. Baseline study procedures for the ART initiation cohort are similar to the SaDAPT trial, without TB investigations for participants without TB symptoms. Data is collected on the same study tablet using the same REDCap instance as for the SaDAPT trial. Participants are followed-up passively, through collection of data available in routine registers and lab reports via register reviews, with the possibility of an in-person assessment together with the routine ART refill visit six months after enrolment. Participants of the SaDAPT trial will continue to be followed-up as part of the ART initiation cohort after termination of the trial after six months. Passive follow-up via register review may continue up to two years after enrolment. Analysis will be by descriptive statistics. The ART initiation cohort is recruiting concurrently with the SaDAPT trial and will close recruitment at the time when the final SaDAPT trial participant is recruited. We estimate the prevalence of any TB symptom among PWH (re)initiating ART to be between 30%-45% [23], thus between 1300 and 1800 participants are expected to be included in the ART initiation cohort.

## Discussion

SaDAPT is a pragmatic randomized non-inferiority trial conducted in Malawi and Lesotho that compares the effectiveness and safety of SDI of ART ("ART first") to ART initiation only after results of TB investigations are available ("TB results first") in PWH with presumptive TB.

A previous trial assessing the effectiveness of SDI in presumptive TB has provided same-day NAAT results as part of the trial procedures and then started ART only in participants found to be TB negative [46]. However, this turnaround time for NAAT is not feasible in many settings. Another trial (SLATE II) allowed SDI for people with TB symptoms only if symptoms were mild and after a negative TB LAM result [25]. Despite the relative simplicity of the SLATE II screening algorithm, implementation in routine practice might be a considerable obstacle because a standardized severity assessment of TB symptoms requires specific training of involved healthcare staff and TB LAM availability is often limited in routine practice. In SaDAPT, the eligibility screening is reduced to the routine assessments required before ART initiation in every non-pregnant patient, as only patients with potential CNS infection and those requiring hospital admission or referral to a higher-level facility due to severe illness are excluded. Hence, we expect the results of SaDAPT to be highly generalizable and relevant to practice guidelines in settings applying a public health approach to ART.

Limitations of the pragmatic approach of the trial are related to the incomplete standardization of procedures between study sites, for example the investigations to rule out or confirm TB are depending on clinical judgement, local guidelines and local availability of diagnostic tools. All study sites have NAAT routinely available, but turn-around times are unpredictable and other tools such as X-ray or urine-LAM are available at some sites but not at others.

One of the main drivers determining the outcome of the trial as well as HIV outcomes in routine care is attrition from care. Therefore, we minimize the trial's influence on routine practices by avoiding additional study visits, with the exception of the primary endpoint assessment six months after enrolment, which falls together with the routine viral load measurement. This implies that we are relying on passive surveillance of adverse events. Therefore, we are only capturing events leading to patient-initiated contact with the healthcare system or reporting to the study team. Hence, the SaDAPT trial does not allow to reliably capture non-severe adverse events, such as mild IRIS cases.

SDI of ART offers considerable advantages for providers and patients compared to ART initiation only after TB results are available if found to be safe and non-inferior. For providers, allowing SDI for patients with presumptive TB would simplify the ART initiation processes and allow for harmonization of initiation procedures independent of the presence of TB symptoms. For patients, the number of clinic visits and hence loss of time and money could be considerably reduced if ART could be safely dispensed on the day of first presentation.

Combining the data collection for the SaDAPT trial with the observational ART initiation cohort that includes patients (re)initiating ART at the study facilities independent of the presence of TB symptoms will enhance generalizability of the SaDAPT trial results and allow for a deeper understanding of the findings in the context of the overall retention and viral suppression rates at the study facilities. Also, the comparison of safety outcomes such as the incidence of TB-IRIS, active TB or SAEs will allow us to check plausibility of the results found in the SaDAPT trial population.

## Supporting information

**S1 Checklist. Clinical study protocol SaDAPT trial.**
(PDF)

**S1 File. SPIRIT checklist.**
(DOC)

## Acknowledgments

We would like to thank all study staff for the dedicated work.

## Author Contributions

**Conceptualization:** Felix Gerber, Alain Amstutz, Peter MacPherson, Rachael Burke, Niklaus Labhardt.

**Funding acquisition:** Felix Gerber, Rachael Burke, Niklaus Labhardt.

**Investigation:** Felix Gerber, Robina Semphere, Blaise Lukau, Palesa Mahlatsi, Timeo Mtenga, Tristan Lee, Maurus Kohler, Mamello Molatelle, Nthuseng Bridgett Marake, Rachael Burke, Niklaus Labhardt.

**Methodology:** Felix Gerber, Tracy Renée Glass, Alain Amstutz, Peter MacPherson, Rachael Burke, Niklaus Labhardt.

**Project administration:** Felix Gerber, Robina Semphere, Blaise Lukau, Palesa Mahlatsi, Timeo Mtenga, Maurus Kohler, Mamello Molatelle, Marriot Nliwasa, Irene Ayakaka, Rachael Burke, Niklaus Labhardt.

**Software:** Tristan Lee, Rachael Burke.

**Supervision:** Peter MacPherson, Marriot Nliwasa, Irene Ayakaka, Rachael Burke, Niklaus Labhardt.

**Writing – original draft:** Felix Gerber, Tracy Renée Glass.

**Writing – review & editing:** Felix Gerber, Robina Semphere, Blaise Lukau, Palesa Mahlatsi, Timeo Mtenga, Tristan Lee, Maurus Kohler, Tracy Renée Glass, Alain Amstutz, Mamello Molatelle, Peter MacPherson, Nthuseng Bridgett Marake, Marriot Nliwasa, Irene Ayakaka, Rachael Burke, Niklaus Labhardt.

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
