## [Decision Letter · Decision Letter 0]

3 Nov 2023

PONE-D-23-20412Same-day versus rapid ART initiation in HIV-positive individuals presenting with symptoms of tuberculosis: protocol for an open-label randomized non-inferiority trial in Lesotho and MalawiPLOS ONE

Dear Dr. Gerber,

Thank you for submitting your manuscript to PLOS ONE. After careful consideration, we feel that it has merit but does not fully meet PLOS ONE’s publication criteria as it currently stands. Therefore, we invite you to submit a revised version of the manuscript that addresses the points raised during the review process.

**Comments raised by 2 reviewers:**

Please provide more information on the rational and justification for a 10% non-inferiority mar**gin.**

**Additional comments from Reviewer 1**

1) “Logistic regression models the probability of an outcome, such as the probability of HIV viral suppression. That may be in line with your goal of comparing proportions (line 230) but it does not agree with testing for difference in risk (line 235). Odds ratios are known to overestimate risk (e.g., https://doi.org/10.1136/bmj.316.7136.989) so if you want to compare risk differences you will need to switch to a different statistical method, such as a log binomial or Poisson model (with a modification to obtain appropriate confidence intervals; see https://doi.org/10.1093/aje/kwh090). Otherwise, the statement of risk difference will need to be changed to match the logistic model. The same approach is mentioned for secondary endpoints as well, so please modify the subsequent paragraph to match any changes.”

2) (lines 156-157) Please provide more information on the randomization process. For instance, what specific or range of block sizes are used? Were there any algorithms used or software packages used?

**Additional points raised by reviewer 2**

1) Explain better the rationale for conducting the study on both people who were ART naïve and with ART interruption

2) Consider to introduce a stratified randomization or sensitivity analysis stratified by ART naïve vs. ART interrupted.

3) incorporate more details on trial intervention and describe the heterogeneity of HIV care practices, particularly for ART initiation.

4) the authors should consider adding more detail about how they will handle OIs, in particular in how intracranial infection was ruled out during study enrollment.

5) A better description of the standard for defining and tracing lost to follow-up patients is needed

We look forward to receiving your revised manuscript.

Kind regards,

Omar Sued, MD, PhD

Academic Editor

PLOS ONE

Journal Requirements:

Additional Editor Comments:

Comments raised by 2 reviewers:

Please provide more information on the rational and justification for a 10% non-inferiority margin.

Additional comments

Adapted from Reviewer 1

1) “Logistic regression models the probability of an outcome, such as the probability of HIV viral suppression. That may be in line with your goal of comparing proportions (line 230) but it does not agree with testing for difference in risk (line 235). Odds ratios are known to overestimate risk (e.g., https://doi.org/10.1136/bmj.316.7136.989) so if you want to compare risk differences you will need to switch to a different statistical method, such as a log binomial or Poisson model (with a modification to obtain appropriate confidence intervals; see https://doi.org/10.1093/aje/kwh090). Otherwise, the statement of risk difference will need to be changed to match the logistic model. The same approach is mentioned for secondary endpoints as well, so please modify the subsequent paragraph to match any changes.”

2) (lines 156-157) Please provide more information on the randomization process. For instance, what specific or range of block sizes are used? Were there any algorithms used or software packages used?

Adapted from reviewer 2

1) Explain better the rationale for conducting the study on both people who were ART naïve and with ART interruption

2) Consider to introduce a stratified randomization or sensitivity analysis stratified by ART naïve vs. ART interrupted.

3) incorporate more details on trial intervention and describe the heterogeneity of HIV care practices, particularly for ART initiation.

4) the authors should consider adding more detail about how they will handle OIs, in particular in how intracranial infection was ruled out during study enrollment.

5) A better description of the standard for defining and tracing lost to follow-up patients is needed

Reviewers' comments:

Reviewer's Responses to Questions

**Comments to the Author**

1. Does the manuscript provide a valid rationale for the proposed study, with clearly identified and justified research questions?

Reviewer #1: Yes

Reviewer #2: Yes

2. Is the protocol technically sound and planned in a manner that will lead to a meaningful outcome and allow testing the stated hypotheses?

Reviewer #1: Yes

Reviewer #2: Yes

3. Is the methodology feasible and described in sufficient detail to allow the work to be replicable?

Reviewer #1: Yes

Reviewer #2: Yes

4. Have the authors described where all data underlying the findings will be made available when the study is complete?

Reviewer #1: Yes

Reviewer #2: Yes

5. Is the manuscript presented in an intelligible fashion and written in standard English?

Reviewer #1: Yes

Reviewer #2: Yes

6. Review Comments to the Author

You may also provide optional suggestions and comments to authors that they might find helpful in planning their study.

Reviewer #1: General comments:

I think this protocol is generally very well written and provides a good justification for the study. The methodology appears feasible to me. The SPIRIT checklist is provided and the trial is registered.

I have a few minor comments in the specific comments section below. My biggest concern that I wanted to touch on here regards the choice of statistical method. Logistic regression models the probability of an outcome, such as the probability of HIV viral suppression. That may be in line with your goal of comparing proportions (line 230) but it does not agree with testing for difference in risk (line 235). Odds ratios are known to overestimate risk (e.g., https://doi.org/10.1136/bmj.316.7136.989) so if you want to compare risk differences you will need to switch to a different statistical method, such as a log binomial or Poisson model (with a modification to obtain appropriate confidence intervals; see https://doi.org/10.1093/aje/kwh090). Otherwise, the statement of risk difference will need to be changed to match the logistic model. The same approach is mentioned for secondary endpoints as well, so please modify the subsequent paragraph to match any changes.

Specific comments:

1. (lines 156-157) I think it would be good to provide a little more information on the randomization process. For instance, what specific or range of block sizes are used? Were there any algorithms used or software packages used?

2. (lines 174, 175, 202) "is" should be "are". I didn't notice any other subject verb disagreement, but I may have missed something. it might be worth just checking this over before resubmitting since PLOS ONE does not copyedit accepted manuscripts.

3. Somewhere in the manuscript, I think it would be good to provide more information on why a 10% non-inferiority margin is appropriate. In any non-inferiority test, the choice of margin is key, so I think some justification for this choice is really important.

Reviewer #2: The findings of this study will be impactful in providing clarity to the HIV care provider communities following the WHO guidelines to timely and safely start people living with HIV on treatment, bringing out both personal and public health benefits. The study has clinical equipoise to conduct a randomized controlled trial, and the study methods is appropriate and feasible. Several suggestions for minor revision and study planning:

- One of the eligibility criteria is ART naïve or interruption for 90 days or more. I am concerned whether people who have interrupted their ART could have developed drug resistance, which might result in failure to achieve viral suppression (primary endpoint). If drug resistance test needed to be conducted before ART initiation, it could result in delayed in ART initiation and affect time to ART initiation (secondary endpoint). The authors should clarify whether the drug resistance test was conducted and provide the rationale for conducting the study on both people who were ART naïve and with ART interruption. The authors should also consider conducting a sensitivity analysis stratified by ART naïve vs. ART interrupted.

- While I recognize that the research team did not intervene in routine care and the procedures were not standardized between study sites, the authors should incorporate more details on trial intervention and describe the heterogeneity of HIV care practices, particularly for ART initiation, which is currently missing in the manuscript. The authors should also clarify what they meant by routine ART initiation procedure, as “routine” can be interpreted differently by different readers, especially since differentiated service models are common in Malawi and Lesotho. Additionally, the authors should consider adding more detail about how intracranial infection was ruled out during study enrollment.

- The author mentioned in line 182 that “the study team … documents the routine procedures and results.” The authors should consider disseminating these documents in future research finding publications. Sharing the components of the trial interventions and how they were implemented will provide readily available information to help other providers/implementers adapt the trial interventions to their settings.

- In line 219, the authors described the tracing procedure. The authors should include how many attempts the research team would make to trace the participants before classifying them as being lost to follow-up.

- In line 235, the authors should provide the rationale behind using non-inferiority margin of 10%.

7. PLOS authors have the option to publish the peer review history of their article (what does this mean?). If published, this will include your full peer review and any attached files.

Reviewer #1: No

Reviewer #2: No

---

## [Author Response · Author response to Decision Letter 0]

19 Dec 2023

Comment raised by both reviewers: 

1. Please provide more information on the rationale and justification for a 10% non-inferiority margin.

o Thank you for raising this important question. It was an oversight to omit this information in the manuscript.

o The non-inferiority margin of 10% was chosen based on clinical reasoning, guideline recom-mendations and previous studies in the field. The US Food and Drug Administration recom-mends a margin of 10-12% for comparing the anchor drugs in treatment-naïve patients in HIV treatment trials (U.S. Department of Health and Human Services, Food and Drug Administra-tion, Center for Drug Evaluation and Research (CDER), 2015). This is also the margin that was used by the majority of HIV trials in the past (Hill & Sabin, 2008). Most importantly, a margin of 10% represents a clinically reasonable minimal effect that should be preserved in order to recommend same-day ART initiation as a safe and effective alternative to delaying ART initia-tion until TB results are available considering its advantages of simplifying initiation proce-dures for providers and clients and the potential reduction of pre-ART loss to follow-up. 

o This information has been added in the revised manuscript on lines 251-258

Additional comments from reviewer 1: 

1. Logistic regression models the probability of an outcome, such as the probability of HIV viral sup-pression. That may be in line with your goal of comparing proportions (line 230) but it does not agree with testing for difference in risk (line 235). Odds ratios are known to overestimate risk (e.g., https://doi.org/10.1136/bmj.316.7136.989) so if you want to compare risk differences you will need to switch to a different statistical method, such as a log binomial or Poisson model (with a modification to obtain appropriate confidence intervals; see https://doi.org/10.1093/aje/kwh090). Otherwise, the statement of risk difference will need to be changed to match the logistic model. The same approach is mentioned for secondary endpoints as well, so please modify the subsequent paragraph to match any changes. 

o Thank you for this important comment. Indeed, there is quite a debate about this topic in the literature (for example: (Wells, 2022)). We agree though that particularly in the case where odds ratios will not be reported, it is more appropriate to estimate the risk differ-ences from a modified poisson model. We have adapted the text accordingly for both pri-mary and secondary outcomes (see lines 245-247 in revised manuscript).

2. (lines 156-157) Please provide more information on the randomization process. For instance, what specific or range of block sizes are used? Were there any algorithms used or software packages used?

o Randomization was done using the ralloc command in Stata version 16.1 and with randomly varying block sizes of 4 and 6. This information was added in the revised manuscript on lines 161-162

3. (lines 174, 175, 202) "is" should be "are". I didn't notice any other subject verb disagreement, but I may have missed something. it might be worth just checking this over before resubmitting since PLOS ONE does not copyedit accepted manuscripts.

o Thank you for pointing out, the suggested changes were made and the manuscript checked again (lines 185, 186, 212 in the revised manuscript)

Additional comments raised by reviewer 2: 

1. Explain better the rationale for conducting the study on both people who were ART naïve and with ART interruption

o From a primary healthcare perspective, both, ART naïve and ART experienced people after treatment interruption, pose similar challenges. Both are at risk of developing IRIS after ART (re)initiation especially in presence of an untreated TB infection. The clinical management in-cluding TB screening and TB investigations, ART (re)initiation, screening for and prevention of other opportunistic infections is the same for both groups. Often people presenting for ART initiation are not disclosing previous treatment episodes, therefore distinguishing between ART naïve and ART experienced people is often not possible in clinical routine. The key ques-tion of SaDAPT whether ART should be (re)initiated on the same day or whether TB should be investigated first is of equal relevance to both groups. SaDAPT is aiming to provide evidence to inform ART programs following a public health approach. Based on these considerations, excluding ART experienced people after treatment interruption would severely limit the gen-eralizability of our results to the relevant clinical setting. 

o This rationale has been added in the revised manuscript on lines 119-125

2. Consider to introduce a stratified randomization or sensitivity analysis stratified by ART naïve vs. ART interrupted

o Thank you for this valuable suggestion. We have added a planned subgroup analysis stratified by ART naïve versus ART interrupted in the revised manuscript on lines 264-265

3. Incorporate more details on trial intervention and describe the heterogeneity of HIV care practices, particularly for ART initiation. The authors should also clarify what they meant by routine ART initia-tion procedure, as “routine” can be interpreted differently by different readers, especially since differ-entiated service models are common in Malawi and Lesotho.

o Further details about the trial intervention and the difference in HIV care practices among the different study sites have been added to the revised manuscript on lines 174-179

4. The authors should consider adding more detail about how they will handle OIs, in particular in how intracranial infection was ruled out during study enrollment.

o The management of opportunistic infections during the study follow-up will not be influenced by study participation but remain fully at the discretion of the routine clinical staff (see lines 174-178 in the revised manuscript). 

o The screening for intracranial infection at enrollment is designed to mimic routine clinical care. As part of the eligibility screening, study staff screens for signs and symptoms suggestive of intracranial infection (i.e. neck rigidity, severe headache, confusion). Presence of any of these signs or symptoms prohibits trial participation. Furthermore, the CD4 value is assessed during screening. If a CD4 value is available and below 200 cells/mm3, enrolment into the trial is only possible after a negative CrAg test. If no CD4 value is available at the time of screening, a clinical assessment of the general health status is required. If a potential participant is se-verely sick or unable to walk unaided, enrolment into the trial requires a negative CrAg result. If a CD4 value equal or above 200 cells/mm3 is available or if an individual is not severely sick in case no CD4 results are available at the time of screening, no CrAg test is required. All clini-cal assessments during screening will be conducted by registered nurses or clinical officers fol-lowing the usual care at the site. (see lines 128-135 of the revised manuscript)

5. In line 219, the authors described the tracing procedure. The authors should include how many at-tempts the research team would make to trace the participants before classifying them as being lost to follow-up. A better description of the standard for defining and tracing lost to follow-up patients is needed.

o Participants will be traced if 30 weeks after enrolment, no viral load has been measured in the endpoint window (22-30 weeks after enrolment) and no death or withdrawal is documented. 

o The tracing entails a minimum of three phone calls on different days to all contact numbers provided by the participant at enrolment. In case of unsuccessful phone calls, one home visit will be conducted. 

o This information was added to the revised manuscript on lines 229-240

6. One of the eligibility criteria is ART naïve or interruption for 90 days or more. I am concerned whether people who have interrupted their ART could have developed drug resistance, which might result in failure to achieve viral suppression (primary endpoint). If drug resistance test needed to be conducted before ART initiation, it could result in delayed ART initiation and affect time to ART initiation (sec-ondary endpoint). The authors should clarify whether the drug resistance test was conducted. 

o No drug resistance testing is conducted at enrolment as baseline drug resistance testing is cur-rently not recommended by local and WHO guidelines and therefore not part of routine pro-cedures. 

7. The author mentioned in line 182 that “the study team … documents the routine procedures and re-sults.” The authors should consider disseminating these documents in future research finding publica-tions. Sharing the components of the trial interventions and how they were implemented will provide readily available information to help other providers/implementers adapt the trial interventions to their settings.

o Thank you for this valuable suggestion. Alongside the trial results, we will also disseminate observation of routine care practices and outcomes. This is one of the major goals of the ART initiation cohort sub-study described in the revised manuscript on lines 307-335

1. Please ensure that your manuscript meets PLOS ONE's style requirements, including those for file nam-ing.

o Adherence to all style requirements is confirmed. 

2. In your Data Availability statement, you have not specified where the minimal data set underlying the results described in your manuscript can be found. PLOS defines a study's minimal data set as the un-derlying data used to reach the conclusions drawn in the manuscript and any additional data required to replicate the reported study findings in their entirety. All PLOS journals require that the minimal da-ta set be made fully available. For more information about our data policy, please see http://journals.plos.org/plosone/s/data-availability. Upon re-submitting your revised manuscript, please upload your study’s minimal underlying data set as either Supporting Information files or to a stable, public repository and include the relevant URLs, DOIs, or accession numbers within your revised cover letter. For a list of acceptable repositories, please see http://journals.plos.org/plosone/s/data-availability#loc-recommended-repositories. Any potentially identifying patient information must be fully anonymized. Important: If there are ethical or legal restrictions to sharing your data publicly, please explain these restrictions in detail. Please see our guidelines for more information on what we consider unacceptable restrictions to publicly sharing data: http://journals.plos.org/plosone/s/data-availability#loc-unacceptable-data-access-restrictions. Note that it is not acceptable for the authors to be the sole named individuals responsible for ensuring data access. We will update your Data Availa-bility statement to reflect the information you provide in your cover letter.

o As outlined in lines 376-380 of the revised manuscript, we will make an anonymized data set available along with publication of the study results. The currently submitted study protocol manuscript does not contain data, hence no data can be made available currently. 

3. Please review your reference list to ensure that it is complete and correct. If you have cited papers that have been retracted, please include the rationale for doing so in the manuscript text, or remove these references and replace them with relevant current references. Any changes to the reference list should be mentioned in the rebuttal letter that accompanies your revised manuscript. If you need to cite a re-tracted article, indicate the article’s retracted status in the References list and also include a citation and full reference for the retraction notice.

o The reference list has been reviewed. We are not citing any retracted article.

---

## [Editor Report · Decision Letter 1]

10 Jan 2024

Same-day versus rapid ART initiation in HIV-positive individuals presenting with symptoms of tuberculosis: protocol for an open-label randomized non-inferiority trial in Lesotho and Malawi

PONE-D-23-20412R1

Dear Dr. Gerber,

We’re pleased to inform you that your manuscript has been judged scientifically suitable for publication and will be formally accepted for publication once it meets all outstanding technical requirements.

Kind regards,

Omar Sued, MD, PhD

Academic Editor

PLOS ONE
---

## [Editor Report · Acceptance letter]

31 Jan 2024

PONE-D-23-20412R1 

PLOS ONE

Dear Dr. Gerber, 

I'm pleased to inform you that your manuscript has been deemed suitable for publication in PLOS ONE. Congratulations! Your manuscript is now being handed over to our production team.

Kind regards, 

on behalf of

Dr. Omar Sued 

Academic Editor

PLOS ONE